# Excitation Wavelength and Colloids Concentration-Dependent Nonlinear Optical Properties of Silver Nanoparticles Synthesized by Laser Ablation

**DOI:** 10.3390/ma15207348

**Published:** 2022-10-20

**Authors:** Tarek Mohamed, Majed H. El-Motlak, Samar Mamdouh, Mohamed Ashour, Hanan Ahmed, Hamza Qayyum, Alaa Mahmoud

**Affiliations:** 1Laser Institute for Research and Applications LIRA, Beni-Suef University, Beni-Suef 62511, Egypt; 2Department of Engineering, Faculty of Advanced Technology and Multidiscipline, Universitas Airlangga, Gubeng 60115, Indonesia; 3Al Anbar Health Directorate, Training and Human Development Centre, Al-Anbar 31001, Iraq; 4High Institute of Optics Technology HIOT, Sheraton Heliopolis, Cairo 11799/5, Egypt; 5Department of Physics, COMSATS University Islamabad, Park Road, Islamabad 45550, Pakistan

**Keywords:** silver nanoparticle, laser ablation, nonlinear optical properties, Z-scan

## Abstract

We reported experimental results from investigations that employed the Z-scan method to explore the dependence of silver nanoparticles’ (AgNPs) nonlinear optical properties on the excitation wavelength, AgNP concentration, and size. Using a 532 nm Nd: YAG laser beam at 100 mJ/pulse for different ablation times, AgNPs were synthesized from a silver target immersed in distilled water. UV–Vis spectroscopy and an atomic absorption spectrometer are used to characterize the optical properties of laser-synthesized AgNPs as well as their concentrations. The AgNPs’ size and shape are determined using a transmission electron microscope (TEM). The laser-synthesized AgNPs are spherical, with an average particle size of 12 to 13.2 nm. Whatever the ablation time, the AgNP colloids exhibit reversed saturable absorption and a negative nonlinear refractive index (n_2_). Both n_2_ and the nonlinear absorption coefficient (α_3_) increase as the AgNP concentration increases. As the excitation wavelength and average size of the AgNPs increase, n_2_ and α_3_ decrease.

## 1. Introduction

Following the invention of high-power lasers, the field of nonlinear optics (NLO) has opened a broad window in science and technology [1]. Higher-order nonlinear properties can become relevant when the electric fields associated with high-intensity laser beams are powerful enough ($>10^{8}\;V/m$) [1,2,3,4]. One of the most important NLO research requirements is the development of novel materials with useful features. Noble metal nanoparticles display high nonlinear optical characteristics around particular wavelengths, making them potentially suitable for photonic devices among a wide range of materials [5,6,7,8,9,10,11,12,13]. Because silver nanoparticles (AgNPs) exhibit a high surface plasmon resonance (SPR) in visible light, they have recently received much attention in various photonic devices [14]. SPR can lead to increased electric fields in a localized region, altering the optical characteristics that could be beneficial in various applications such as optical limiting, electronics, and optical signal processing [15,16]. Among various optical characteristics, nonlinear absorption is considered the most important because it can alter strong light propagation in the medium and lead to various applications [11].

There are several methods for studying the effects of nonlinear optical properties, such as the spatial beam distortion [17], nonlinear interferometry [18], two-wave interferometry [19], four-wave interferometry [16], and Z-scan method [19,20]. However, some of these techniques necessitate elaborate experimental setups and expensive equipment. As a result of its relative simplicity, precision, and sensitivity, the Z-scan approach established by Sheik-Bahae et al. [17] has become a basic instrument for determining the nonlinear properties of the material [21,22].

Although there have been a few reports on the investigation of NLO properties of AgNPs using the Z-scan approach [23,24,25], no systematic study of the NLO of AgNP colloids employing femtosecond laser pulses at different concentrations and excitation wavelengths has been undertaken to the best of our knowledge. In [23], a femtosecond laser pulse with an excitation wavelength of 800 nm was used to investigate the NLO characteristics of AgNPs. They observed that the size of AgNPs has an effect on their nonlinear optical properties. The NLO properties of AgNPs were studied using ps pulses from a Nd: YAG laser at a wavelength of 532 nm [24]. They showed that the AgNP colloidal solution had a negative nonlinear refractive index and a reverse saturable absorption coefficient (self-defocusing). The nonlinear refractive index and nonlinear absorbance of AgNPs in aqueous solutions were investigated in [25], and it was observed that the nonlinear response is dependent on nanoparticle concentration.

Laser ablation in liquids (LAL) is one of the foremost vital physical techniques that provide a promising method for manufacturing NPs from its bulk form. In LAL, the high-intensity laser light is targeted on the surface of the solid target immersed in liquid [26,27]. The size and shape of nanoparticles created by LAL can be controlled by adjusting the laser beam parameters [28]. Compared with other approaches for obtaining metal colloids, LAL is advantageous due to its capability to obtain a higher purity of the nanomaterial with an easier, faster, and more direct methodology that does not involve many steps or long process times. Moreover, LAL is an adaptable and environmentally friendly methodology that enables the synthesis of various styles of NPs without the utilization of any toxic or venturesome substances. Laser ablation is a one-step process that can result in ready-to-use batches without the need for additional sorting procedures [29].

In this study, we presented detailed experimental studies on the NLO properties of AgNPs using the Z-scan technique with 100 fs laser pulses at constant excitation power of 1 W and different excitation wavelengths, concentrations, and AgNP sizes. Pulsed laser ablation in the liquid technique was used to synthesize AgNPs. The laser-synthesized AgNPs were characterized using UV–Vis spectroscopy and a transmission electron microscope (TEM). The Z-scan technique investigated the NLO properties of laser-synthesized AgNPs for various excitation wavelengths ranging from 740 to 820 nm. Furthermore, the influence of AgNP concentrations and sizes on their NLO properties was studied.

## 2. Material and Methods

### 2.1. Laser Ablation Setup

Figure 1 shows a schematic diagram of the laser ablation setup. A laser ablation technique was used to synthesize AgNPs from a bulk silver target placed in distilled water. The bulk silver has a purity of 99.99 percent and a rectangular shape of 25×15×1.9 mm. Before laser ablation, the Ag target was cleaned to remove the oxide surface that had formed due to air exposure. The Ag target was placed in a glass beaker with 50 mL of distilled water after being held by a holder. The Ag target was immersed in a thin layer of distilled water (about 3 mm thick) and was subjected to laser irradiation through that layer. To protect the convex lens from water splashing, the top of the beaker was sealed with a glass cover with a hole comparable in size with the laser beam [30]. The Ag target was ablated using an Nd: YAG (Quanta-Ray PRO 350, Irvine, CA, USA) laser system with a 10 ns pulse duration and a 10 Hz repetition rate. The laser energy per pulse was set at 100 mJ, and the irradiation time ranged from 5 to 15 min. A magnetic stirrer was used during the ablation to uniformly distribute the synthesized AgNPs in the solution and move the particles away from the laser beam direction, where they absorb the laser beam and reduce ablation efficiency.

### 2.2. Z-Scan Setup

The Z-scan setup [31] is depicted in Figure 2. The AgNP colloidal samples were subjected to an intense tunable pulsed femtosecond laser (INSPIRES HF100) from spectra physics (Mountain View, CA, USA). With a repetition rate of 80 MHz, this system can cover a wavelength range from 345 to 2500 nm. The used femtosecond laser was pumped by a mode-locked femtosecond Ti: Sapphire laser (MAI-TAI HP) from spectra physics, which has a wavelength range from 690 nm to 1040 nm and a repetition rate of 80 MHz. The spatial profile of the pulsed laser beam is Gaussian, with a spatial mode of TEM_00_ and an M^2^ factor < 1.1. Using horizontally polarized laser pulses with a 100 fs duration, the nonlinear optical response of AgNPs colloids was examined. A quartz cuvette glass (from Hellma Analytics, Plainview, NY, USA) with a 1 mm optical path was employed to fill with AgNPs colloids. The input Gaussian laser beam was focused using a convex lens with a 5 cm focal length to a beam waist of 17.2 μm, which was measured by the knife-edge method. The total transmitted intensity from the sample is measured as a function of sample location relative to the focus using the optical power meter (Newport 843–R). The transmittance “S” of the aperture is adjusted to be S = 0.3 and S = 1 correspondingly for both closed aperture (CA) as well as open aperture (OA) measurements.

## 3. Results and Discussions

### 3.1. Characterization of AgNPs Colloids

One of the most effective approaches for determining the optical response of metallic nanoparticles is UV–Vis spectroscopy. The UV–Vis absorption spectra of silver colloids were measured by a spectrophotometer (Model: Branson UV-1510, wavelength accuracy 0.1 nm, Branson Ultrasonics Corp., Danbury, CT, USA). This technique is useful due to colloidal metal nanoparticles’ high surface plasmon resonances [32]. The sharp SPR peak around the 400 nm region corresponds to colloidal AgNPs [33,34]. Several factors can influence the SPR peak position, such as the shape, size, and concentration of AgNPs. Figure 3 illustrates UV–Vis absorption spectra of colloidal AgNPs obtained at different ablation periods of 5, 10, and 15 min with a 100 mJ/pulse ablation energy. The SPR peaks for samples prepared for 5, 10, and 15 min were found at 399, 398.5, and 397.5 nm, respectively. The shift of the peak position toward a shorter wavelength with increasing laser ablation time can be attributed to the synthesis of smaller particle sizes with increasing laser ablation time. It was also found that as ablation time increased, the SPR peaks became sharper as in the case of 15 min ablation time. The FWHMs of SPR peaks were 86.56, 69.85, and 68.34 nm for 5, 10, and 15 min ablation times, respectively. This decrease in the FWHM of the peak can be due to the reduction in the particle size distribution with increasing laser ablation time. The average size and size distribution of AgNPs were verified using a transmission electron microscope (TEM). Figure 4a–c depict the histogram of the size distribution of AgNPs produced by ablation with a laser in distilled water at various ablation times. The inset in Figure 4 shows the TEM micrographs of the AgNPs, which shows that AgNPs are mostly spherical with an average size of 12–13.2 nm. When the laser ablation time was increased from 5 to 15 min, the size of the AgNPs decreased from 13.2 nm to 12 nm. The laser fragmentation of AgNPs could explain the decreases in NP size with increasing ablation time [35]. When the ablation time was increased, the already synthesized AgNPs were fragmented into smaller NPs, resulting in increased concentration of AgNPs. Figure 5 displays the Tauc’s plot of AgNP colloids at various ablation times. The optical bandgap ($E_{g}$) measurements for the AgNP colloids are obtained using Tauc’s equation for the direct bandgap of ($\alpha =\frac{\sqrt{A\;\left(E-E_{g}\right)}}{E}$) [36], where A is the proportionality constant, $E_{g}$ is the energy bandgap, α is the absorption coefficient, and E is the incident photon energy. The direct bandgap was investigated by extrapolating the straight line of Tauc’s plot (2(α E)=0) as shown in Figure 5. The relation between the energy bandgap and size of AgNP colloids is depicted in Figure 6. The bandgap energy increases as the size of AgNPs increases. Using an atomic absorption spectrometer, the concentration of AgNP colloids was measured at different ablation times (Agilent Technologies, Santa Clara, CA, USA, 200 Series AA). Figure 7 shows how the size and concentration of AgNPs are directly proportional to the ablation time. As ablation time increases, the size of AgNPs decreases, while concentration increases.

### 3.2. Open Aperture Z-Scan

As previously indicated, the direct bandgap of AgNP colloids ranges from 5.01 to 5.21 eV, demonstrating that three-photon absorption is sufficient to excite the samples in the wavelength range from 740 to 820 nm. This wavelength range was chosen because it is far away from AgNP surface plasmon resonance and hence avoids significant linear absorption, as illustrated in Figure 3. The open aperture (OA) Z-scan results for the AgNP colloids at concentrations of 3.35, 6.74, and 7.38 mg/L are shown in Figure 8a–c when the incoming excitation power was 1 W and the excitation wavelengths ranged from 740 to 820 nm. The dots indicate the experimental data, whereas the solid line represents the fitting using Equation (1). All of the acquired curves express a “valley” as a result of the reverse saturable absorption (RSA) effect (positive nonlinear absorption) that is consistent with the previous literature [37,38]. The nonlinear absorption can be caused by a variety of processes in nanoparticles, including free-carrier absorption [33], nonlinear scattering [39,40], intraband and interband transitions [33], and three-photon absorption [41]. Nonlinear scattering is another factor that contributes to RSA. The temperature of AgNPs rises when Ag atoms absorb laser energy, resulting in the formation of rapidly expanding microplasmas. Microbubbles are formed by the rapid generation of microplasmas and the diffusion of heat energy throughout the fluid medium. The incident laser beam is scattered by microbubbles and microplasmas, leading to RSA [41]. The three-photon nonlinear absorption coefficient (*α*_3_) was extracted from the best fit of the experimental data shown in Figure 8a–c using the following equation [20,42]:(1)$$\Delta T_{OA}=1-\frac{\alpha_{3}I_{0}^{2}L}{3^{\frac{3}{2}}{\left[1+{\left(\frac{Z}{Z_{o}}\right)}^{2}\right]}^{2}}$$
where Δ*T_OA_* is an expression for normalized transmission, α_3_ is the three-photon nonlinear absorption coefficient, *I_0_* is the maximal intensity at the focus (*Z* = 0), and *L* and *Z*_0_ are the sample thickness and Rayleigh length, respectively. The obtained value of α_3_ at various concentrations of AgNP colloids as a function of excitation wavelengths is shown in Figure 9. As the excitation wavelength decreases from 820 to 740 nm, the α_3_ increases, which could be attributed to increased free-carrier absorption [43].

### 3.3. Closed Aperture Z-Scan

The closed aperture (CA) Z-scan method is used to determine the nonlinear refractive index of AgNP colloids. In this work, the value of CA was configured so that it transmits a fixed percentage of 30% of the laser beam that was passed through the OA setting. CA Z-scan studies were performed for excitation wavelengths ranging from 740 to 820 nm and with an excitation power of 1 W.

Because the light traveling across the sample is partially absorbed, heat accumulates, and the sample does not reach its equilibrium temperature. Acoustic waves are generated due to this heat that alters the density of the medium, resulting in a variation in the refractive index [44]. This change in the refractive index with temperature change is known as the thermal lens effect. In the thermal lens effect mechanism, the fluctuation in the refractive index closely matches the spatial beam profile. The thermal effect appears when the time interval of laser pulses is shorter or close to the characteristic thermal time (*t_c_* = w^2^/4D) [45], where w is the beam waist and D is the thermal diffusion coefficient. In the case of liquid, *t_c_* is 40 μs [44], which is much larger than the 12.5 ns (1/80 MHz) separation time among consecutive laser pulses. This causes heat accumulation in the sample, preventing it from returning to equilibrium in the interval between pulses. As a result, it is clear that the thermal lens significantly impacts the refractive index n_2_ determination. A repetition rate of 1 kHz is often enough to create thermal lensing in the experiment [46]. Moreover, not only repetition rate but also the pulse duration can contribute to ambiguous results [46]. When evaluating the thermal lens effects on the refractive index, a recently established technique [47] compensates for the accumulative thermal lensing in samples produced by a high-repetition-rate laser pulse. The thermal focal length is calculated using the following equation:(2)1f(Z)=aLEPfL32ω2(Z)(1−1NP)
where $a=\alpha \left(dn/dT\right)/2k{\left(\pi^{3}D\right)}^{1/2}$ is regarded as a parameter for fitting, *L* is the thickness of the sample, *E_P_* is the energy per laser pulse, *f_L_* is the repetition rate, *N_P_* is the number of laser pulses, and *ω^2^*(*Z*) is the beam radius at the sample. The following equation is used to calculate the normalized transmittance of the CA Z-scan, which is mostly determined by the thermal focal length [48]:(3)$$\Delta T_{CA}=1-\frac{2Z}{f\left(Z\right)}$$
where *Z* indicates the sample’s position; the nonlinear refractive index can be determined using the following equation [47,48,49]:(4)$$n_{2}=\frac{\lambda \omega_{o}^{2}\Delta \varphi }{2P_{peak}L_{eff}}$$
where *P_peak_* is the peak power and *L_eff_* is the effective lens of the sample given by $L_{eff}=\left(1-e^{-n\gamma L}\right)/n\gamma$, where *γ* is the linear absorption and *n* = 1 or 2 represents two- or three-photon absorption, respectively; Δ*φ* is the on-axis phase shift determined by
(5)$$\Delta \varphi =\frac{Z_{o}}{2f\left(0\right)}$$
where f(0) represents the focal length of the induced thermal lens once the sample is positioned at Z = 0.

The CA Z-scans for different concentrations of 3.35, 6.74, and 7.38 mg/L are shown in Figure 10a–c, respectively. The solid black curves in Figure 10 were fitted to experimental data with *L* = 1 mm, *E_P_* = 12.5 nJ, *f_L_* = 80 MHz, and *N_P_* = 19.2 × 10^9^ pulses using Equations (2) and (3). Changing only parameter a in Equation (2), which is material dependent, obtained the best fit to the closed aperture experimental data. The results show that transmissions have a prefocal peak followed by a postfocal valley, regardless of concentrations. This result suggests that AgNPs have a negative nonlinear refractive index. These findings are consistent with previously reported literature [50]. Furthermore, the CA experimental results show that increasing the excitation wavelength causes a decrease in the normalized transmittance peak-to-valley difference ΔT_P-V_.

At various AgNP concentrations, Figure 11 depicts the nonlinear refractive index n_2_ as a function of the laser excitation wavelength. Figure 11 shows that, due to the influence of SPR, the n_2_ linearly decreases as the laser excitation wavelength increases for all AgNP colloid samples. The nonlinear refractive index decreases as the excitation wavelength is far from the SPR [51]. Furthermore, as the AgNP concentrations increase, n_2_ increases, which is owing to the AgNP size decreasing.

### 3.4. Effect of Concentration on NLO Properties of AgNP Colloids

Figure 12 shows the influence of AgNP colloid concentration on the three-photon absorption coefficient and nonlinear refractive index at 1 W excitation power and 800 nm excitation wavelength. As the concentration of AgNPs decreases, the value of both the nonlinear refractive index and nonlinear absorption coefficient decreases too due to the influence of concentration. It is obvious that when small NPs fit into a small volume, this leads to the decrease in the fraction volume and therefore the decrease in the nonlinear optical coefficients. These results were consistent with previous studies [52]. However, this is not consistent with Ref. [53], as the nonlinear optical properties of nanoparticles depend on the features, shape, size, and method of synthesis of the nanoparticles as well as the excitation laser parameters.

The dependence of the nonlinear refractive index and nonlinear absorption coefficient on the average AgNP size is shown in Figure 13. As demonstrated in Figure 13, the nonlinear refractive index and nonlinear absorption coefficient are inversely proportional to the average size of AgNPs. This inverse relationship arises because as nanoparticle sizes increase, a small number of nanoparticles can fit into the same volume, resulting in a decrease in the fraction volume and, as a result, a decrease in the nonlinear refractive index and nonlinear absorption coefficient values.

### 3.5. Three-Photon Absorption Cross-Section

The three-photon absorption cross-section (3PACS) of AgNP colloids can be determined using the following relationship derived from the measured nonlinear absorption coefficient *α*_3_ [54]:(6)$$\sigma_{3}^{\prime }=\frac{\alpha_{3}}{N_{A}\;d_{0}10^{-3}}{\left(\frac{hc}{\lambda }\right)}^{2}$$
where *N_A_* stands for Avogadro’s constant, $d_{0}$ (mol/L) is the concentration, and $\left(\frac{hc}{\lambda }\right)\;$ is the photon energy. Figure 14 illustrates the dependence of the three-photon absorption cross-section on the excitation wavelength when the excitation power was kept constant at 1 W. With increasing excitation wavelength, the three-photon absorption cross-section decreases. This is because when the excitation wavelength decreases, the free carrier absorption increases, resulting in an increase in the three-photon absorption cross-section. Moreover, a decrease in 3PACS was also observed with the increasing concentration of the AgNPs due to particle agglomeration.

## 4. Conclusions

Using a 532 nm Nd: YAG laser, different concentrations of AgNP colloids (3.35, 6.74, and 7.38 mg/L) were synthesized by pulsed laser ablation in liquid. Transmission electron microscopy and UV–Vis spectroscopy were used to analyze the laser-synthesized AgNPs colloids. The results of the TEM analysis demonstrated that the laser-generated AgNPs are spherical, with a modest decrease in the AgNP size as the laser ablation time was increased. At ablation times of 5, 10, and 15 min, the average AgNP sizes were 13.2, 13, and 12 nm, respectively. The UV–Vis spectroscopy revealed an increase in the bandgap with increasing AgNP size. The OA and CA Z-scan methods were used to investigate the nonlinear optical properties of AgNP colloids at a constant excitation power of 1 W, different concentrations of AgNP colloids, different AgNP sizes, and different excitation wavelengths. As the concentration of AgNPs rises, both the nonlinear refractive index (n_2_) and nonlinear absorption coefficient (α_3_) rise due to the influence of the average size. n_2_ and α_3_ were inversely proportional to the AgNP average size. The AgNP colloids show reversed saturable absorption and a negative nonlinear refractive index regardless of the ablation time. n_2_ and α_3_ decreased as the excitation wavelength increased. At a concentration of 3.35 mg/L of AgNPs, the maximum values of α_3_ and n_2_ were $5\;\times 10^{-20}\;\left({\mathrm{cm}}^{3}/w^{2}\right)$ and $4.055\;\times 10^{-15\;}\left({\mathrm{cm}}^{2}/w\right)$, respectively, at an excitation wavelength of 740 nm.

## Figures and Tables

**Figure 1 materials-15-07348-f001:**
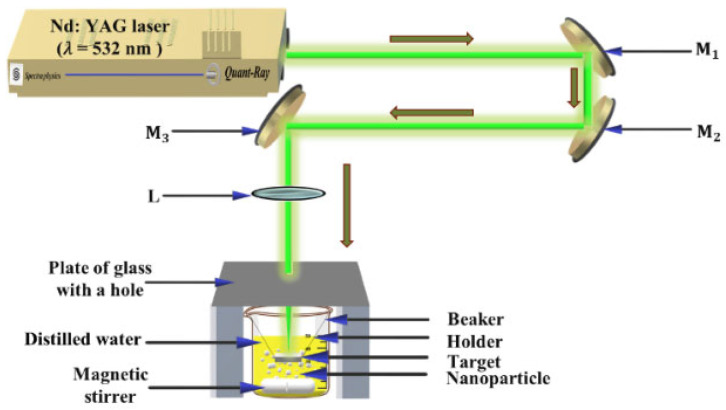
Schematic diagram of laser ablation experimental setup. L, convex lens; M1, M2, and M3, highly reflecting mirrors.

**Figure 2 materials-15-07348-f002:**
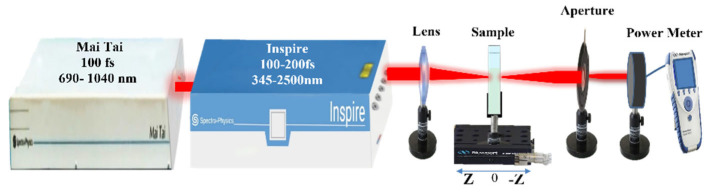
Setup for Z-scan experiment.

**Figure 3 materials-15-07348-f003:**
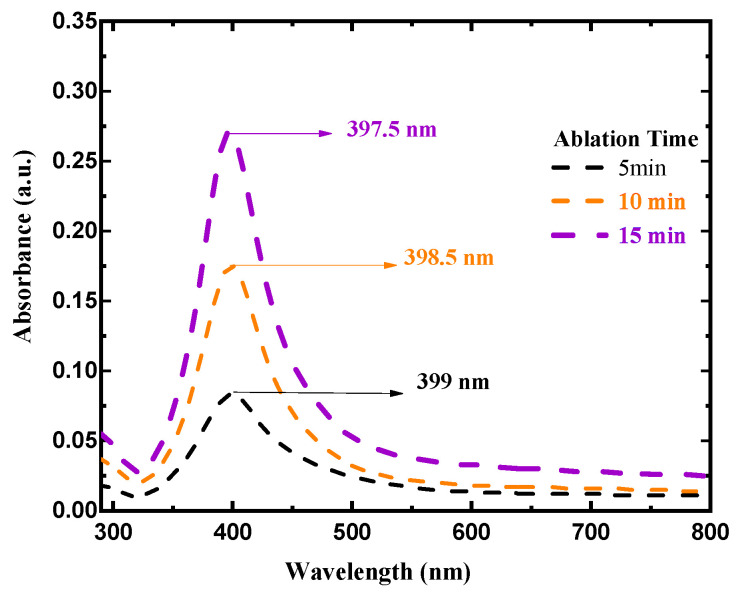
UV–Vis absorbance spectra of AgNP colloids at constant ablation energy of 100 mJ/pulse and at different ablation times.

**Figure 4 materials-15-07348-f004:**
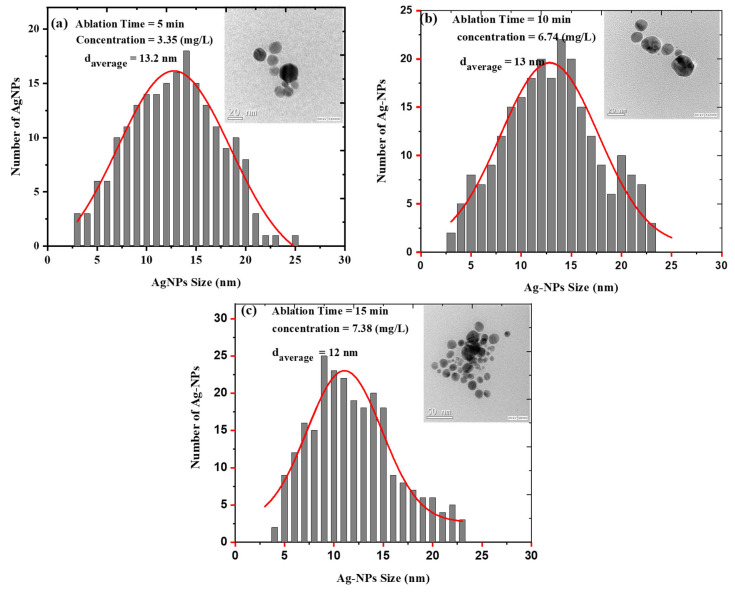
Size distribution of AgNPs synthesized at various ablation times. Insets show TEM images of AgNPs. (**a**) 5 min laser ablation time (**b**) 10 min laser ablation time (**c**) 15 min laser ablation time.

**Figure 5 materials-15-07348-f005:**
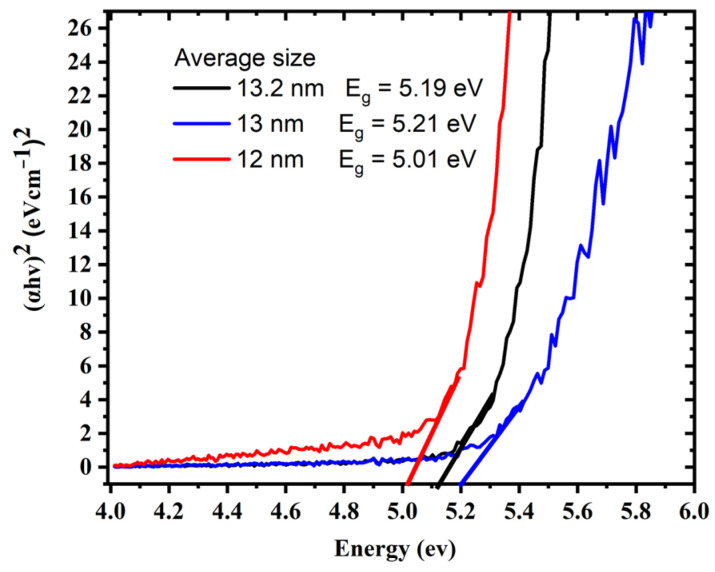
Tauc’s plot extrapolation to determine the bandgap of AgNP colloids for different ablation times.

**Figure 6 materials-15-07348-f006:**
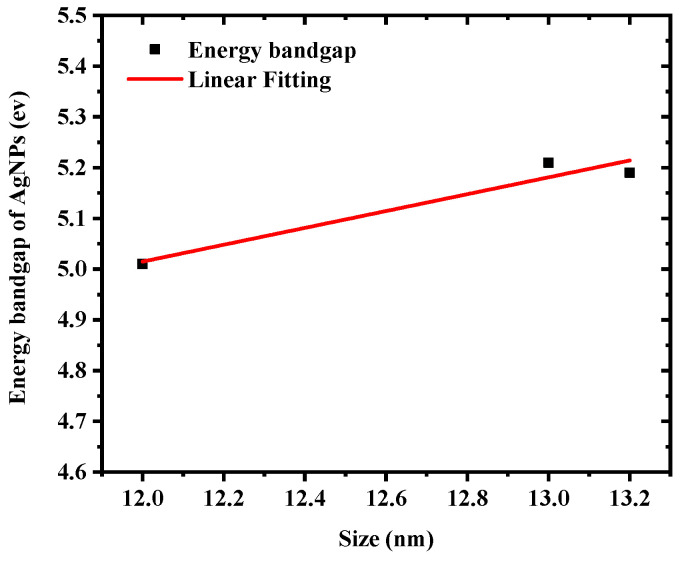
Dependence of bandgap energy on average size of AgNP colloids.

**Figure 7 materials-15-07348-f007:**
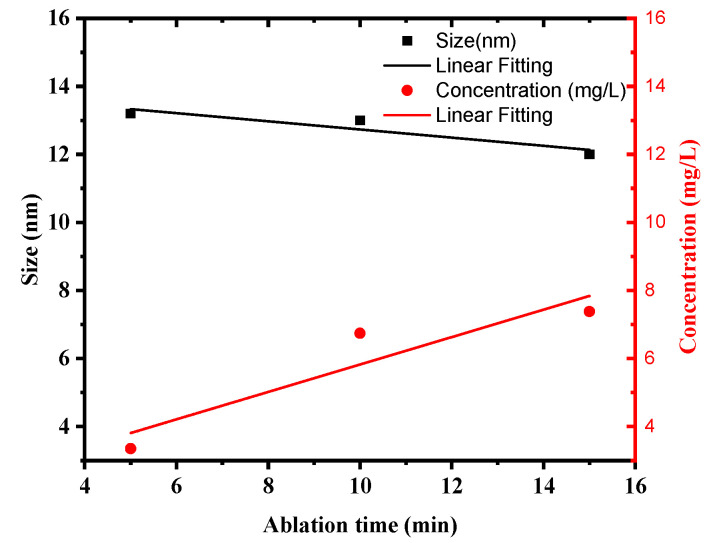
Dependence of AgNP concentration and size on ablation time.

**Figure 8 materials-15-07348-f008:**
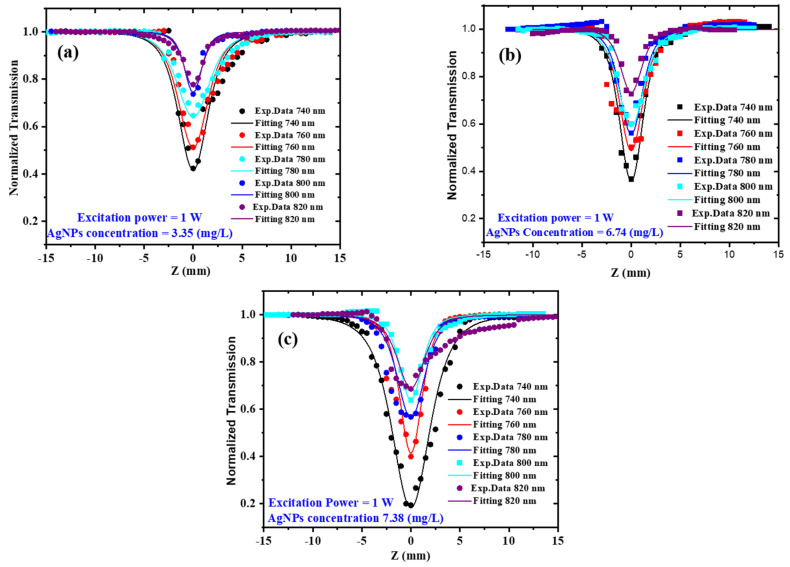
OA Z-scan measurements for AgNP colloids at constant excitation power of 1 W, different excitation wavelengths ranging from 740 to 820 nm, and at different AgNP concentrations of (**a**) 3.35 mg/L, (**b**) 6.74 mg/L, and (**c**) 7.38 mg/L.

**Figure 9 materials-15-07348-f009:**
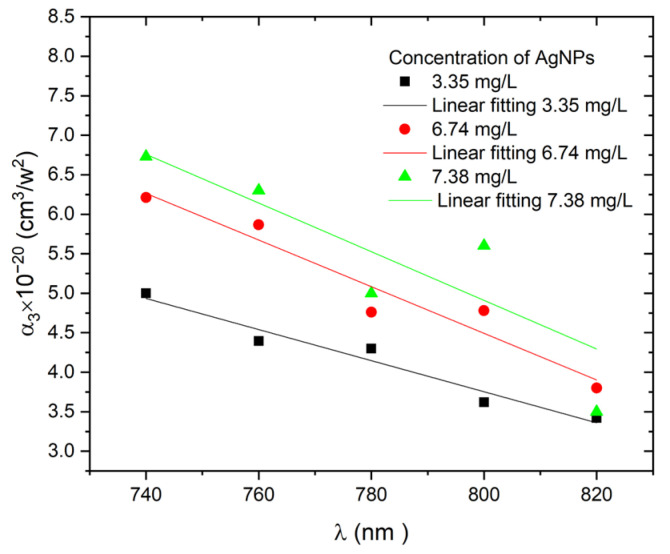
Dependence of *α_3_* on excitation wavelengths at constant AgNP concentrations and at constant excitation power of 1 W.

**Figure 10 materials-15-07348-f010:**
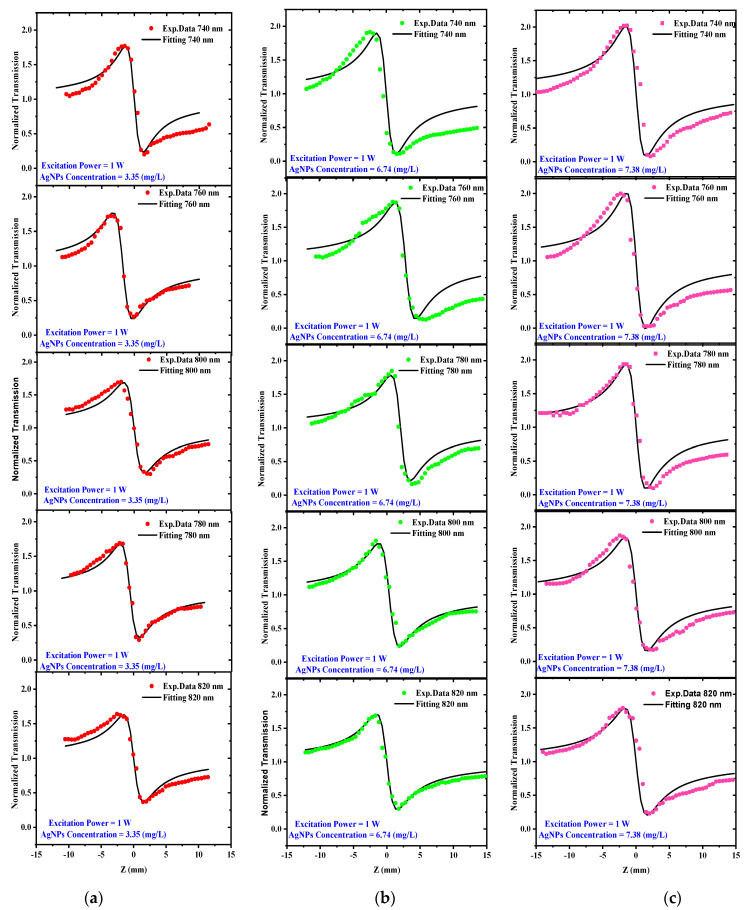
CA Z–scan transmission for AgNPs at 1 W excitation power, different excitation wavelengths between 740 and 820 nm, and various AgNP concentrations of (**a**) 3.35 (mg/L), (**b**) 6.74 (mg/L), and (**c**) 7.38 (mg/L).

**Figure 11 materials-15-07348-f011:**
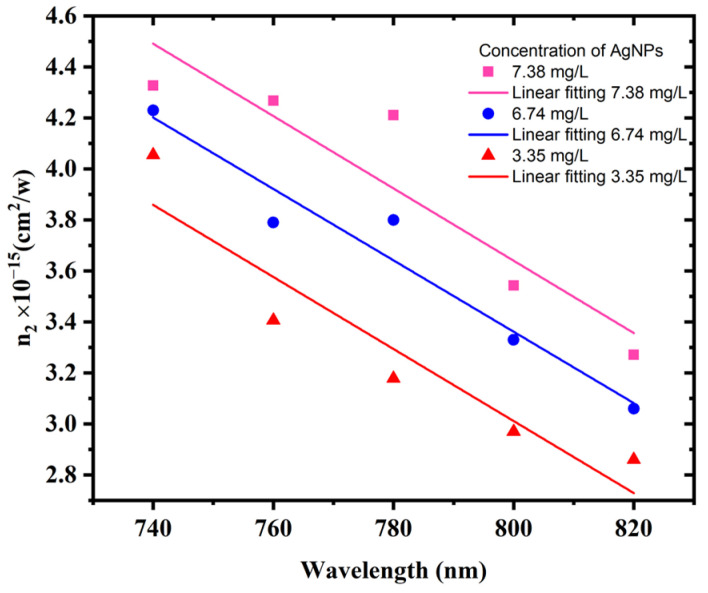
Variation of n_2_ as a function of excitation wavelength at different AgNP concentrations.

**Figure 12 materials-15-07348-f012:**
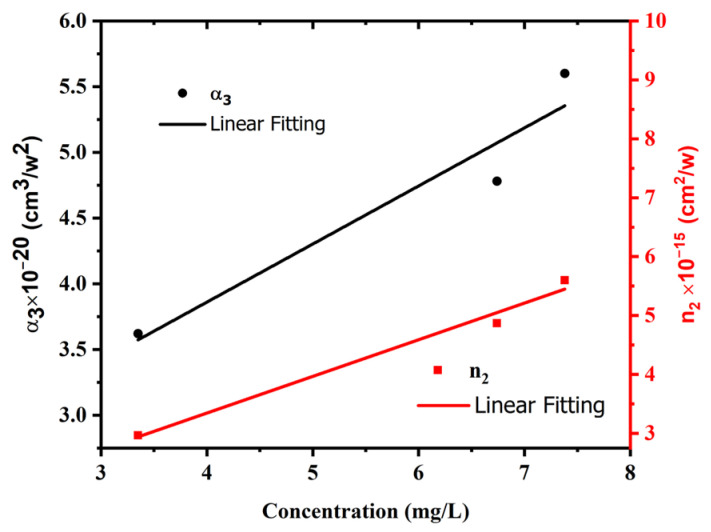
Dependence of *α_3_* and n_2_ on AgNP colloid concentration at constant excitation power and wavelength of 1 W and 800 nm, respectively.

**Figure 13 materials-15-07348-f013:**
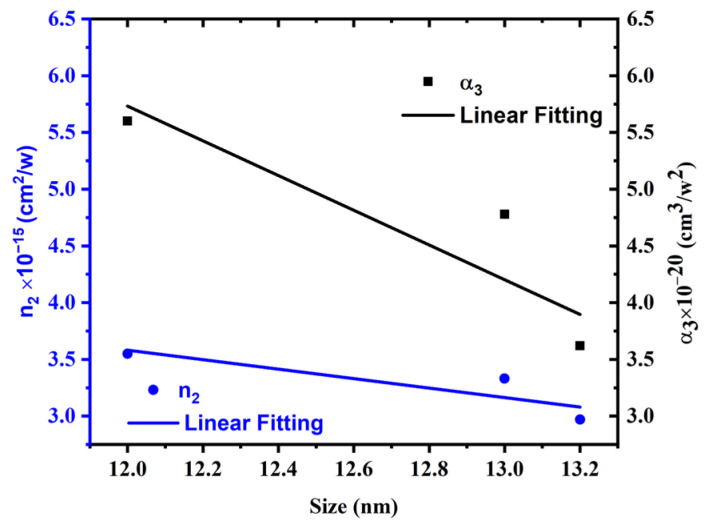
Dependence of *n*_2_ and *α*_3_ on average particle size at constant excitation power and wavelength of 1 W and 800 nm, respectively.

**Figure 14 materials-15-07348-f014:**
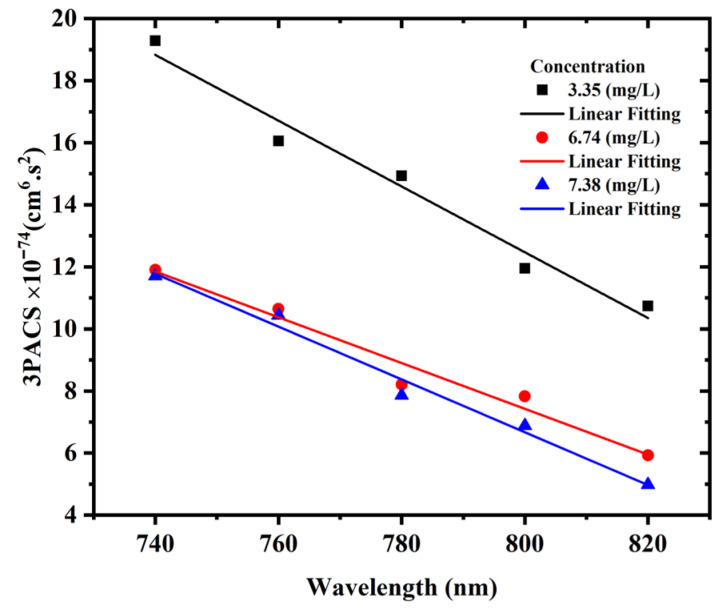
Dependency of three-photon absorption cross-section of AgNP colloids on excitation wavelength at constant AgNP concentrations and constant excitation power of 1 W.

## Data Availability

Data underlying the results presented in this paper are not publicly available at this time but may be obtained from the authors upon reasonable request.

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
