# Peer review of "Excitation Wavelength and Colloids Concentration-Dependent Nonlinear Optical Properties of Silver Nanoparticles Synthesized by Laser Ablation"

_materials, 2022, doi:10.3390/ma15207348_

Round 1
Author Response
Response to Reviewer 1 Comments
In the paper “Excitation Wavelength and Colloids Concentration Dependent Nonlinear Optical Properties of Silver Nanoparticles Synthesized by Laser Ablation” the authors report Ag nanoparticle synthesis and characterization. Nanoparticles are synthesized using Laser Ablation and their nonlinear optical properties are characterized using the Z-scan method. This work has novel results, but the following comments need to be addressed:
We'd like to thank the reviewer for providing such positive feedback on the manuscript.
Point 1: Regarding literature, authors should look also at these papers regarding what type of measurements have been carried out for silver nanoparticles (10.1364/JOSAB.440997; 10.3952/physics.v55i2.3100).
Response 1: We appreciate the reviewer's remarks. The recommended articles are excellent resources, so we have included them in the reference list.
Point 2: Regarding the Z-scan setup, the authors present the beam size at the focal point (17.2 um). Was it measured using some beam profile measurements or calculated?
Response 2: We appreciate the reviewer's input. The knife edge approach was used to determine the size of the beam. The revised manuscript has been modified in response to the reviewer's feedback.
Point 3: When analyzing nonlinear absorption, did the authors consider the possibility of 2PA effects due to the SPR effect? When processing open-aperture data, the authors could show if the data better fit the two-photon process or three-photon process using specific analytical models. Also, power-dependent measurements should be included to show for sure that the process is a three-photon process. At the current stage, there is not enough analytical proof that this really is a three-photon process. Power dependence would be the most concrete proof of that.
Response 3: Thank you for the valuable comments.
First, regarding the relationship between the possibility that the 2PA is due to the SPR. The open aperture measurements were done for excitation wavelengths ranging from 740 to 820 nm (far from the SPR region). In this range, 3PA is comparable to the energy gap and fits the experimental data better than 2PA.
Second, to determine whether the experimental open aperture data fit better with 3PA or 2PA, the OA experimental data at excitation wavelength 820nm and excitation power of 1 W was fitted with 3PA and 2PA, respectively. The graph below illustrates that 3PA better represents the experimental data than 2PA.
Third, the reported results were obtained by investigating the dependency of NLO of AgNPs colloids on excitation wavelengths and concentrations. These results confirm that 3PA better represents the experimental data than 2PA. However, excitation power dependence was not investigated in the study as reported. The reviewer's comments are crucial for following up on the work that has been presented and may be applied to our upcoming study.
Point 4: Regarding closed-aperture measurements, it would also be beneficial to analyze data using a standard Z-scan analytical model and show what is the ratio of peak-valley distance compared to Rayleigh length. This would be a way to prove that thermal effects are responsible for observed nonlinearities. Also, a decrease in nonlinear refractive index with an increase in wavelength should be commented on. Does this correlate with linear absorption if those are thermal effects?
Response 4: We appreciate the reviewer's feedback. The ordinary model of the Z-scan (Ref. 42) doesn’t handle the thermal effect caused by the high repetition rate excitation fs laser. To avoid this issue and in the current manuscript, we used a modified model (Ref. 47) which takes into account the thermal effect due of using a high repetition rate excitation fs laser. This model (Ref. 47) was developed by our group as a new method for deducing the nonlinear refractive index of media when excited by a high repetition rate fs laser pulses. It is based on solving the heat equation to derive an expression for the normalized transmittance of closed-aperture Z-scan measurements as a function of the accumulative thermal lens induced by a high repetition rate fs laser pulses. Using this method allows us to find a better adjustment between the experimental and theoretical Z-scan curves in the far field.
Point 5: In section 3.4 authors write “As the concentration of AgNPs rises, the value of both the nonlinear refractive index and the nonlinear absorption coefficient rises due to the influence of average size.” and “As demonstrated in Fig. 13, the nonlinear refractive index and nonlinear absorption coefficient are inversely proportional to the average size of AgNPs. This inverse relationship arises because as nanoparticle sizes increase, a small number of nanoparticles can fit into the same volume, resulting in a decrease in fraction volume and, as a result, a decrease in nonlinear refractive index and nonlinear absorption coefficient values”. Is there any way to separate the concentration and size influence? The authors refer to reference 49 which gives an opposite relation between NP size and nonlinear refractive index (in that paper it increases). If the authors give this paper as a reference this discrepancy should be addressed. Is it possible that the size difference between samples is too small and all effect comes only from concentration? More analysis on this should be given.
Response 5: We appreciate the feedback provided by the reviewer. A brief discussion has been included in the revised version of the manuscript to address these concerns.
Point 6: Also, 3PACS analysis should be expanded especially regarding concentration dependence. Can an explanation be given why smaller concentration leads to a higher value? Here also power dependent measurements would help as it would show if some saturation of this effect is observed.
Response 6: We'd like to thank the reviewer for their valuable feedback. The revised version of the manuscript now includes a brief discussion that addresses the concentration dependence. As reported, the study did not investigate the excitation power dependency. The reviewer's remarks are very important for following up on the work that has been presented and could be used in our forthcoming study.

Reviewer 2 Report
Review
In the paper “Excitation Wavelength and Colloids Concentration Dependent Nonlinear Optical Properties of Silver Nanoparticles Synthesized by Laser Ablation” the authors report Ag nanoparticle synthesis and characterization. Nanoparticles are synthesized using Laser Ablation and their nonlinear optical properties are characterized using the Z-scan method. This work has novel results, but the following comments need to be addressed:
1. Regarding literature, authors should look also at these papers regarding what type of measurements have been carried out for silver nanoparticles (10.1364/JOSAB.440997; 10.3952/physics.v55i2.3100),
2. Regarding the Z-scan setup, the authors present the beam size at the focal point (17.2 um). Was it measured using some beam profile measurements or calculated?
3. When analyzing nonlinear absorption, did the authors consider the possibility of 2PA effects due to the SPR effect? When processing open-aperture data, the authors could show if the data better fit the two-photon process or three-photon process using specific analytical models. Also, power-dependent measurements should be included to show for sure that the process is a three-photon process. At the current stage, there is not enough analytical proof that this really is a three-photon process. Power dependence would be the most concrete proof of that.
4. Regarding closed-aperture measurements it would also be beneficial to analyze data using a standard Z-scan analytical model and show what is the ratio of peak-valley distance compared to Rayleigh length. This would be a way to prove that thermal effects are responsible for observed nonlinearities. Also, a decrease in nonlinear refractive index with an increase in wavelength should be commented on. Does this correlate with linear absorption if those are thermal effects?
5. In section 3.4 authors write “As the concentration of AgNPs rises, the value of both the nonlinear refractive index and the nonlinear absorption coefficient rises due to the influence of average size.” and “As demonstrated in Fig. 13, the nonlinear refractive index and nonlinear absorption coefficient are inversely proportional to the average size of AgNPs. This inverse relationship arises because as nanoparticle sizes increase, a small number of nanoparticles can fit into the same volume, resulting in a decrease in fraction volume and, as a result, a decrease in nonlinear refractive index and nonlinear absorption coefficient values.”. Is there any way to separate the concentration and size influence? The authors refer to reference 49 which gives an opposite relation between NP size and nonlinear refractive index (in that paper it increases). If the authors give this paper as a reference this discrepancy should be addressed. Is it possible that the size difference between samples is too small and all effect comes only from concentration? More analysis on this should be given.
6. Also 3PACS analysis should be expanded especially regarding concentration dependence. Can an explanation be given why smaller concentration leads to a higher value? Here also power dependent measurements would help as it would show if some saturation of this effect is observed.
I recommend this paper for publication after suitable answers has been given to these questions.

Author Response
Response to Reviewer 2 Comments
In this work, authors presented non-linear studies of nanosecond laser ablated Ag nanoparticles in water. Authors have characterized the nanoparticles obtained during the three different time scales. The nanoparticles were well characterized and the z-scan studies were also performed systematically.
However, the experiments performed were not novel and the manuscript lacks novelty. Furthermore, the results obtained were not satisfactorily explained. In my opinion this manuscript is observational manuscript and the results need to be explained with suitable literature. The manuscript is not recommendable to be published in this journal in this scientific level. Here are the concerns to be addressed in the future submissions.
We'd like to thank the reviewer for providing such positive feedback on the manuscript. A brief discussion has been included in the revised version of the manuscript to address these concerns.
Point 1: Line 33: “(˃108 ?/?)” need to be changed as “(˃108 ?/?)” with correct superscript.
Response 1: We appreciate the reviewer comments, which we have now incorporated into the revised version of the manuscript.
Point 2: Line 156: Give a complete description of setup used for obtaining the data for Tauc’s plot.
Response 2: We appreciate the reviewer comments, which we have now incorporated into the revised version of the manuscript.
Point 3: The major concern in the manuscript is the incident power of 1W. Researchers usually used “mW” of power delivered from femtosecond oscillators. Authors need to perform again with low incident powers to make sure and minimize the less thermal effects which are predominant when oscillators are used.
Response 3: Thank you for the valuable comments. At average powers below 1W, the samples' nonlinearity was not observed during the experiment. This has been discussed in the introduction section. Furthermore, we believe that the thermal effect, which has been addressed in most of the previous studies, is mostly caused by the high repetition rate of the excitation laser source.

Reviewer 3 Report
In this work, authors presented non-linear studies of nanosecond laser ablated Ag nanoparticles in water. Authors have characterized the nanoparticles obtained during the three different time scales. The nanoparticles were well characterized and the z-scan studies were also performed systematically.
However, the experiments performed were not novel and the manuscript lacks novelty. Furthermore, the results obtained were not satisfactorily explained. In my opinion this manuscript is observational manuscript and results are need to be explained with suitable literature.The manuscript is not recommendable to be published in this journal in this scientific level. Here are the concerns to be addressed in the future submissions.
1. Line 33: “(˃108 ?/?)” need to be changed as “(˃108 ?/?)” with correct superscript.
2. Line 156: Give a complete description of setup used for obtaining the data for Tauc’s plot.
3. The major concern in the manuscript is the incident power of 1W. Researchers usually used “mW” of power delivered from femtosecond oscillators. Authors need to perform again with low incident powers to make sure and minimize the less thermal effects which are predominant when oscillators are used.
Author Response
Response to Reviewer 3 Comments
In this work, Excitation wavelength and colloids concentration dependent nonlinear optical properties of silver nanoparticles synthesized by laser ablation. The idea of the research is interesting to readers. The background is well studied and the presentation of the method is very clear and sound.
We'd like to thank the reviewer for providing such positive feedback on the manuscript.
Point 1: The author should correct the typo errors in the throughout manuscript.
Response 1: We appreciate the reviewer comments, which we have now incorporated into the revised version of the manuscript.
Point 2: Suitable references should provide in the materials and method section.
Response 2: We appreciate the reviewer comments, which we have now incorporated into the materials and method section of revised version of the manuscript.
Point 3: TEM images of AgNPs should provide more number of particle images.
Response 3: We appreciate the reviewer comments, which we have now incorporated into the revised version of the manuscript.
Point 4: The conclusion section should contain some quantitative information.
Response 3: We appreciate the reviewer comments, which we have now incorporated into the revised version of the manuscript.

Reviewer 4 Report
In this work, Excitation wavelength and colloids concentration dependent nonlinear optical properties of silver nanoparticles synthesized by laser ablation. The idea of the research is interesting to readers. The background is well stidied and the presentation of the method is very clear and sound.
The author should correct the typo errors in the throughout manuscript.
Suitable references should provide in the materials and method section.
TEM images of AgNPs should provide more number of particle images.
The conclusion section should contain some quantitative information.
Author Response
Response to Reviewer 4 Comments
In this work, Excitation wavelength and colloids concentration dependent nonlinear optical properties of silver nanoparticles synthesized by laser ablation. The idea of the research is interesting to readers. The background is well studied and the presentation of the method is very clear and sound.
We'd like to thank the reviewer for providing such positive feedback on the manuscript.
Point 1: The author should correct the typo errors in the throughout manuscript.
Response 1: We appreciate the reviewer comments, which we have now incorporated into the revised version of the manuscript.
Point 2: Suitable references should provide in the materials and method section.
Response 2: We appreciate the reviewer comments, which we have now incorporated into the materials and method section of revised version of the manuscript.
Point 3: TEM images of AgNPs should provide more number of particle images.
Response 3: We appreciate the reviewer comments, which we have now incorporated into the revised version of the manuscript.
Point 4: The conclusion section should contain some quantitative information.
Response 3: We appreciate the reviewer comments, which we have now incorporated into the revised version of the manuscript.

Round 2
Reviewer 2 Report
All my comments have been addressed and the paper can be published in its current state.
Author Response
Response to Reviewer 2 Comments
All my comments have been addressed and the paper can be published in its current state.
We'd like to thank the reviewer for providing such positive feedback on the revised manuscript.
Reviewer 3 Report
The 1 W of incident power from fs oscillator is not required for AgNPs and infact few mW would be sufficient.
Hence, there is some serious flaw in it or authors need to be more careful with measuring power.
Hence this work should not be accepted for publication, in my opinion.
Author Response
Response to Reviewer 3 Comments
Point 1: The 1 W of incident power from fs oscillator is not required for AgNPs and in fact few mW would be sufficient.
Response 1: Thank you for the valuable comments. First of all, we would like to emphasize that the power given (one Watt) is average power, not peak power. Second, we tried this experiment with lower average power initially, but we were unable to detect a nonlinear effect from the AgNPs samples. Third, the 1W is below the sample's damage threshold.
Most of the previous studies that investigated the nonlinear optical properties of metal nanoparticles at different power levels are given in the table below. The current experimental study was conducted at peak excitation intensities ranging from 1.2 to 1.7 MW/cm2, which is less intense than or comparable to the earlier studies.
|
Reference |
Sample |
Peak intensity |
|
[1] |
AuNPs |
55 GW/cm2 |
|
[2] |
Au:Al2O3 Au:SiO2 |
2 × 106 MW/cm2 2 × 106 MW/cm2 |
|
[3] |
Au NPs doped borate |
5.2 × 102 MW/cm2. |
|
[4] |
AgNPs |
1.4 kW/cm2 |
|
[5] |
AgNPs |
10 MW/cm2 |
[1] Zhang, Y. X., & Wang, Y. H. (2017). Nonlinear optical properties of metal nanoparticles: a review. RSC advances, 7(71), 45129-45144.
[2] Ryasnyanskiy, A. I., Palpant, B., Debrus, S., Pal, U., & Stepanov, A. (2007). Third-order nonlinear-optical parameters of gold nanoparticles in different matrices. Journal of luminescence, 127(1), 181-185.
[3] Jagannath, G., Eraiah, B., NagaKrishnakanth, K., & Rao, S. V. (2018). Linear and nonlinear optical properties of gold nanoparticles doped borate glasses. Journal of Non-Crystalline Solids, 482, 160-169.
[4] Zulina, N. A., Pavlovetc, I. M., Baranov, M. A., & Denisyuk, I. Y. (2017). Optical, structural and nonlinear optical properties of laser ablation synthesized Ag nanoparticles and photopolymer nanocomposites based on them. Optics & Laser Technology, 89, 41-45.
[5] Shi, H., Wang, C., Zhou, Y., Jin, K., & Yang, G. (2012). Silver nanoparticles grown in organic solvent PGMEA by pulsed laser ablation and their nonlinear optical properties. Journal of nanoscience and nanotechnology, 12(10), 7896-7902.
